# Chemical disruption of ABA signaling overcomes high-temperature inhibition of seed germination and enhances seed priming responses

James Eckhardt[1,2,3], Aditya Vaidya[1,2,4], Sean Cutler[1,2]*

1 Institute for Integrative Genome Biology, University of California, Riverside, Riverside, CA, United States of America, 2 Botany and Plant Sciences, University of California, Riverside, Riverside, CA, United States of America, 3 HydroGreen, Sioux Falls, SD, United States of America, 4 MilliporeSigma, Temecula, CA, United States of America

* sean.cutler@ucr.edu

**Data Availability Statement:** All data presented can be found in supplementalData.xlsx.

## Abstract

Seed germination is critical to agricultural productivity because low germination rates and/or asynchronous germination negatively affect stand establishment and subsequent yields. Exposure to high temperatures during seed imbibition can decrease both germination synchrony and rates through an ABA-mediated process called thermoinhibition. Methods to reduce thermoinhibition would be agriculturally valuable, particularly with increasing global mean temperatures. Lettuce seed germination is particularly sensitive to high temperatures and is a classic system for studying thermoinhibition. Extensive evidence using mutants and carotenoid biosynthetic inhibitors (*e.g.* fluridone) has demonstrated that endogenous abscisic acid (ABA) biosynthesis is required for thermoinhibition in lettuce and Arabidopsis. Although fluridone and related carotenoid biosynthetic inhibitors block thermoinhibition, they are not well-suited for this application due to their herbicidal effects. Here we explore the potential of ABA receptor antagonism to disrupt thermoinhibition using antabactin (ANT), a broad-spectrum high-affinity receptor antagonist. We show low µM ANT treatments (10 µM) during lettuce seed imbibition reduces thermoinhibition at temperatures of up to 40˚C, demonstrating that ABA signaling is required for thermoinhibition and that receptor antagonists are well-suited anti-thermoinhibition agents. We further explored interactions between ANT and seed priming, which is used commercially to improve seed germination and reduce thermoinhibition and is achieved by partial hydration and subsequent desiccation of seeds. We show that co-priming with ANT improves germination at elevated temperatures better than priming alone, and thus, the two treatments can be combined to improve germination. Our data demonstrate that ABA antagonists are potentially useful agrochemical leads for mitigating the effects of high temperatures on seed germination and stand establishment that may be of increasing importance due to climate change. More generally, ABA antagonists should be useful in physiological processes where ABA's effects are counterproductive to yield.

**Funding:** This work was supported by NSF IOS grant 1656890. There was no additional external funding received for this study.

## Introduction

Seed germination is a vital phase in the life cycle of seed plants, influenced by both developmental and environmental factors. During seed development, the seeds of most plants acquire dormancy, which prevents germination under otherwise favorable conditions [1]. To alleviate dormancy, dry storage (after-ripening) or moist chilling (stratification) are often required [2]. Once dormancy is relieved, seed germination is influenced by environmental factors. For example, in many species, non-dormant seeds exposed to supraoptimal temperatures fail to germinate due to a process called thermoinhibition [3, 4]. As global mean temperatures rise, thermoinhibition is problematic for agriculture as it can cause low or asynchronous germination; developing strategies to mitigate the effects of thermoinhibition are critical, particularly in crop species where synchronous, rapid seed germination is necessary for developmental synchrony, stand establishment and yield.

The plant hormone abscisic acid (ABA) is a key regulator of seed dormancy and germination. Numerous lines of evidence from work in Arabidopsis and lettuce have established that both ABA synthesis and signaling are required for thermoinhibition [5–8]. Lettuce has served as a model system for studying thermoinhibition. QTL studies implicated LsNCED4, a seed-expressed 9-cis-epoxycarotenoid dioxygenase required for ABA biosynthesis, as a determinant of thermotolerance in *L. sativa* [7, 9, 10]. Subsequent analyses of CRISPR-induced LsNCED4 mutations confirmed that NCED4 is required for thermoinhibition [11]. In Arabidopsis, loss of function NCED6 alleles similarly showed increased germination at high temperatures [5, 6]. In both Arabidopsis and lettuce, fluridone treatments, which lower ABA levels, prevent thermoinhibition and enable seed germination at otherwise inhibitory temperatures, demonstrating that *de novo* ABA synthesis is required for thermoinhibition of germination [5, 7, 8]. In addition, multiple ABA-insensitive mutants are less sensitive to thermoinhibition than wild-type Arabidopsis [6]. Thus, the role of ABA in inhibiting seed germination under supraoptimal temperatures is well established.

In the United States, over 95% of lettuce is produced in California and Arizona [12], and thermoinhibition is a persistent problem that reduces emergence, stand establishment, and yield [13, 14]. Currently, the main intervention known to improve germination under supraoptimal temperature is seed priming [14]. Seed priming is a widely used pre-sowing treatment to enhance germination, which involves partially hydrating seeds under controlled conditions to initiate germination and then halting the process prior to radicle emergence. This is achieved through osmotic and/or time-regulated methods, followed by drying and storing the seeds [15, 16]. New methods or treatments to improve seed germination would be of general agricultural utility, given the upward trajectory of global temperatures.

In this work, we investigate if ABA signaling antagonists can be used alone or combined with priming to improve germination under high temperatures, focusing on the newly described ABA antagonist antabactin (ANT) [17]. ABA binds to soluble PYR/PYL/RCAR receptors to trigger a conformational change that allows them to complex with and inhibit clade A PP2C phosphatases; this, in turn, leads to the activation of downstream SnRK2 kinases and various downstream cellular effectors [18]. Antabactin (ANT), is a potent (picomolar $K_d$) pan-receptor ABA receptor antagonist that was designed to disrupt ABA receptor-PP2C interactions and reduce ABA signaling *in planta* [17]. We have previously shown that ANT treatments accelerate Arabidopsis, tomato, and barley seed germination and block Arabidopsis thermoinhibition [17]. ANT additionally blocks ABA-induced transcriptional responses (i.e. Arabidopsis RD29B and MAPKK18 in Arabidopsis). Here, we examine whether ANT can mitigate thermoinhibition in lettuce both on its own or in combination with seed priming. We demonstrate that ANT treatments prevent thermoinhibiton in seeds over a wide temperature

range and can be combined with seed priming to mitigate the inhibitory effects of high temperatures on seed germination.

## Methods

Seeds of *Lactuca sativa* 'Salinas' were grown in a greenhouse in summer 2021 and summer 2022 in Riverside, California. After seeds reached maturity, plants were dried for two weeks and seeds were harvested on August 23, 2021 and August 10, 2022. Seeds for priming experiments were harvested on August 10, 2022, October 1, 2022 and January 20, 2023. Sungro Professional Growing Mix was mixed with fertilizer (Osmocote) and insecticide (Marathon 1% Granular) according to manufacturers' instructions. Plants were fully dried and seeds were collected and after ripened at room temperature in the dark for 1–5 months before experiments were carried out.

### Seed germination assays

**Surface sterilization.** Lettuce seeds were sterilized in room lighting and temperature using the following steps. Seeds were gently shaken for one minute in 70% ethanol. This was followed by fifteen minutes of gentle shaking in 20% bleach and three subsequent washes in sterile water. Seeds were immediately plated after sterilization.

**Antabactin promotion of germination ($EC_{50}$).** To establish the concentration of ANT at which 50% of seeds germinate under thermoinhibition, we conducted assays as follows (adapted from Vaidya et al. 2021). Surface sterilized Salinas seeds (n>35 seeds; seeds were harvested two months prior to the experiment) were plated in Petri dishes on 0.7% agar medium containing ½ Murashige-Skoog (MS) and 10, 5, 1, 0.75, 0.5, 0.25, 0.1 µM ANT and mock untreated control (0.1% v/v DMSO). DMSO was used as a carrier solvent to solvate ANT and was added to the media at a 0.1% v/v ratio. Petri dishes were immediately transferred to 32°C and 25°C incubators in the dark. After 48 hours, plates were photographed. Germination was scored as positive based on radicle emergence, and the $ET_{50}$ (time at which 50% of seeds have germinated) values were inferred by fitting the germination data over time to a 4-parameter log-logistic model in the drc (dose-response curves) R package [19].

**Chemical manipulation of ABA biosynthesis and signaling.** To compare the effect of fluridone, AbamineSG, and ANT on germination under thermoinhibition, we conducted the following assay: surface sterilized Salinas seeds (n>35 seeds; seeds were harvested two months prior to the experiment) were plated in Petri dishes on 0.7% agar medium containing 1/2 MS salts and either 10 µM ANT, 10 µM fluridone, 100 µM AbamineSG or mock untreated control (1% v/v DMSO). Petri dishes were immediately transferred to 32°C and 25°C incubators in the dark. After 48 hours, plates were photographed. Germination was scored upon radicle emergence.

**Comparison with other hormone treatments.** Surface sterilized Salinas seeds (n = 40 seeds) were plated in Petri dishes on 0.7% agar medium containing 1/2 MS salts and either 10 µM ANT, 100 µM GA, 100nM $KAR_1$ or mock untreated control (1% v/v DMSO). Petri dishes were immediately transferred to 32°C and 25°C incubators in the dark. After 48 hours, plates were photographed. Germination was scored upon radicle emergence.

**Effective temperature range of ANT.** We performed the following assay to determine the upper-temperature range at which ANT promotes germination. Surface sterilized Salinas seeds (n>35 seeds; seeds were harvested two months prior to the experiment) were plated in Petri dishes on 0.7% agar medium containing 1/2 MS salts and either 10 µM ANT or mock untreated control (0.1% v/v DMSO). Petri dishes were immediately transferred to 25°C, 32°C, 37°C, 40°C and 45°C incubators in dark. Petri dishes were imaged at 48 hours and 120 hours. After the 120-hour time point, all plates were transferred to the 25°C incubator and imaged

after 72 hours in the dark at 25˚C to determine if the seeds were thermoinhibited or thermo-dormant. Seeds incubated at 45˚C were stained with tetrazolium to determine viability. Staining was carried out as described in HCRM Catão et al. 2018 [20].

## Gene expression assays

Gene expression assays were conducted using RNA isolated from seeds and leaves. Salinas seeds were grown on Petri dishes on 0.7% agar medium containing 1/2 MS salts and either 10 µM ANT or mock untreated control (0.1% v/v DMSO). Seeds were grown for 24 hours in the dark before RNA extraction. Each treatment was conducted in biological triplicate. 50 mg of whole seed tissue was collected and used for RNA extraction. To measure ABA responses in leaf tissue, ~ 1 cm leaf discs were isolated from 2-week-old seedlings and incubated with either 100 µM ABA, 100 µM ABA, and 50 µM ANT or mock untreated control (0.1% v/v DMSO) for 8 hours in 70 µmol/m$^2$/s light before RNA extraction. The total RNA was extracted using the RNeasy Plant Mini Kit (QIAGEN, Cat No. 74904). Immediately after collection, plant tissue was added to QIAGEN Buffer RLC and homogenized using three 10-second cycles of 6500rpm in a Precellys 24 after which extraction was completed as described by QIAGEN. Genomic DNA was eliminated and cDNA was synthesized with the QuantiTect Reverse Transcription Kit (QIAGEN, Cat No. 205311) according to the manufacturer's instructions using 500ng of RNA. Quantitative RT-PCR was performed with CFX Connect Real-Time PCR Detection System using Maxima SYBR Green/Fluorescein qPCR Master Mix (Thermo Scientific, Cat No. K0242) according to the manufacturer's instructions. Initial denaturation at 95˚C was run for 3:00, followed by 40 cycles of denaturation (95˚C for 0:10), annealing (55˚C for 0:10) and extension (72˚C for 0:20). The relative expression level of the genes was normalized against the reference gene UBC21 by ΔCT Method. Statistical tests were performed using the raw ΔCT values. A pairwise t-test was used to assess the differences between treatments.

## The primers used in the experiments are listed as follows

LsUBC21: 5' TCTTAGATCACCGTCCCATCGT3', 5' TCTGAGATTGTCCGAGGATATGAG3'
  LsNCED4: 5'TGATCCAGCGGTTCAGCTAA3', 5'TTCACCAATTACCTCCAGACCAT3'
  LsM3K18A (Lsat_1_v5_gn_7_91420): 5'ATGATTGAAATGGCCACCGG3', 5'TGATCGG
AAAATTCATCTGGGA3'.

## Seed priming

Seeds were primed in -1.25mPa PEG8000 (0.2968g/mL). Water potential was calculated as in Michel 1983 [21]. Seeds were placed in Petri dishes on blotting paper with 10mL of PEG solution and either 10 µM ANT or mock untreated control (0.1% v/v DMSO). Seeds were at 9˚C in low light (1 µM) for 48 hours, rinsed three times with water, dried at 32˚C in an incubator for 2 hours, and then stored for >48 hours at 9˚C in the dark before use. For germination experiments, seeds were surface sterilized and plated in Petri dishes on 0.7% agar medium containing 1/2 MS salts. Seeds were incubated in the dark at 35˚C for 24 hours before germination was quantified.

Approximately 10mg of primed seeds were extracted in 500uL MeOH. Seeds were put through two homogenization cycles in a Precellys 24. Three 10-second cycles of 6500rpm were used. After homogenization, tubes were centrifuged at 13.3rpm for 2 minutes, and 100uL of supernatant was used for LC-MS. For ANT spiked samples, a 1% v/v solution of ANT in MeOH was added to PEG-only primed seeds to make 125, 250, 500, and 1000nM final concentrations. The analysis of ANT was performed on an Agilent 1260 HPLC system coupled with an Agilent 6224 ESI TOF in positive ion mode. Chromatographic separation was carried out

on an Agilent Poroshell 120 EC-C18 column (50 x 3mm, 2.7 μM) using mobile phases A (water/formic acid, 100/0.1, v/v) and B (acetonitrile/formic acid, 100/0.1, v/v). The flow rate was 0.5mL/min and the injection volume was 5uL.

## Phylogeny

The coding regions of the 21 mitogen-activated protein kinase kinase kinases (MAP3K) were downloaded from TAIR. Homologous genes to MAP3K18 in *Lactuca sativa* were identified by using NCBI blastn, and three homologous sequences were selected. Using MEGA (Version 11.0.13), the nucleotide sequences were aligned by MUSCLE [22]. A maximum likelihood tree was then constructed in MEGA using default settings. Bootstrap support values were derived from 500 replicate analyses in MEGA.

# Results

## ANT and germination efficacy

To test if thermoinhibition requires ABA signaling, we tested if ANT treatments could improve seed germination at supra-optimal temperatures, using the ABA biosynthetic inhibitor fluridone as a positive control. We also tested abamineSG, which has not previously been tested in lettuce but in Arabidopsis, which was shown to inhibit the NCEDs required for ABA biosynthesis (see Fig 1C for the structures of the molecules tested). At 32˚C in dark conditions, *L. sativa* 'Salinas' germination was greatly reduced, with mean germination between 0–18% after 48 hours (Fig 1A, 1B. Control seeds, germinated at 25˚C, showed 100% germination, and fluridone (10 μM), a carotenoid biosynthetic inhibitor, fully restored germination (to thermo-inhibited seeds at 32˚C). ANT treatment (10 μM) also restored germination under high temperatures to 97% (Fig 1B). AbamineSG (100 μM), an NCED inhibitor, did not promote germination (Fig 1B). In addition to biosyntheticinhibitors, ANT was compared to common seed treatments (Fig 1D). ANT is more potent than gibberellic acid (GA) and karrikin 1 (KAR$_1$) treatments. GA (100 μM) and KAR$_1$ (100 nM) treatments led to similar germination rates to mock after two days in the dark at 32˚C, 8%, 10% and 18%, respectively. As before, 10 μM ANT nearly fully restored germination (94%, Fig 1D). ANT matches the germination-enhancing capabilities of fluridone and is more effective than GA, KAR1 and abamineSG, underscoring its efficacy in counteracting the adverse effects of high temperatures on seed germination and demonstrating the importance of ABA signaling in thermoinhibition. Collectively, these experiments establish chemical inhibition of ABA signaling can be used to overcome thermoinhibition in lettuce.

To assess the effectiveness of ANT in mitigating thermoinhibition across a range of temperatures and ANT concentrations, we conducted two experiments. The first experiment explored the impact of varying ANT concentrations (0.25–10 μM) on thermoinhibited lettuce seeds. In the second experiment, we investigated the germination rates of ANT and mock-treated seeds at temperatures ranging from 32˚C to 45˚C. ANT is potent in the low micromolar range and at high temperatures. The EC$_{50}$ of ANT is 2.11 μM (Fig 1E). Lettuce seeds were grown for two days in the dark at 32˚C in six ANT concentrations (0.25, 0.5, 1, 2.5, 5, and 10 μM). 10 μM ANT nearly fully restored germination (91% mean), and 0.25 μM ANT had little effect (7% mean). Since 10 μM ANT restored germination under thermoinhibition, we tested to determine the upper-temperature limit of ANT effect. After 120 hours, ANT promoted germination at 37˚C and 40˚C (S2 Fig). At 37˚C, 10 μM ANT treatment resulted in 82% germination, and at 40˚C ANT treatment resulted in 74% germination. Germination was also quantified at 48 hours, and the higher temperatures slowed germination at 37˚C and 40˚C, leading to only 47% and 8% germination, respectively (Fig 1F). At 45˚C, lettuce seeds were dead and no longer

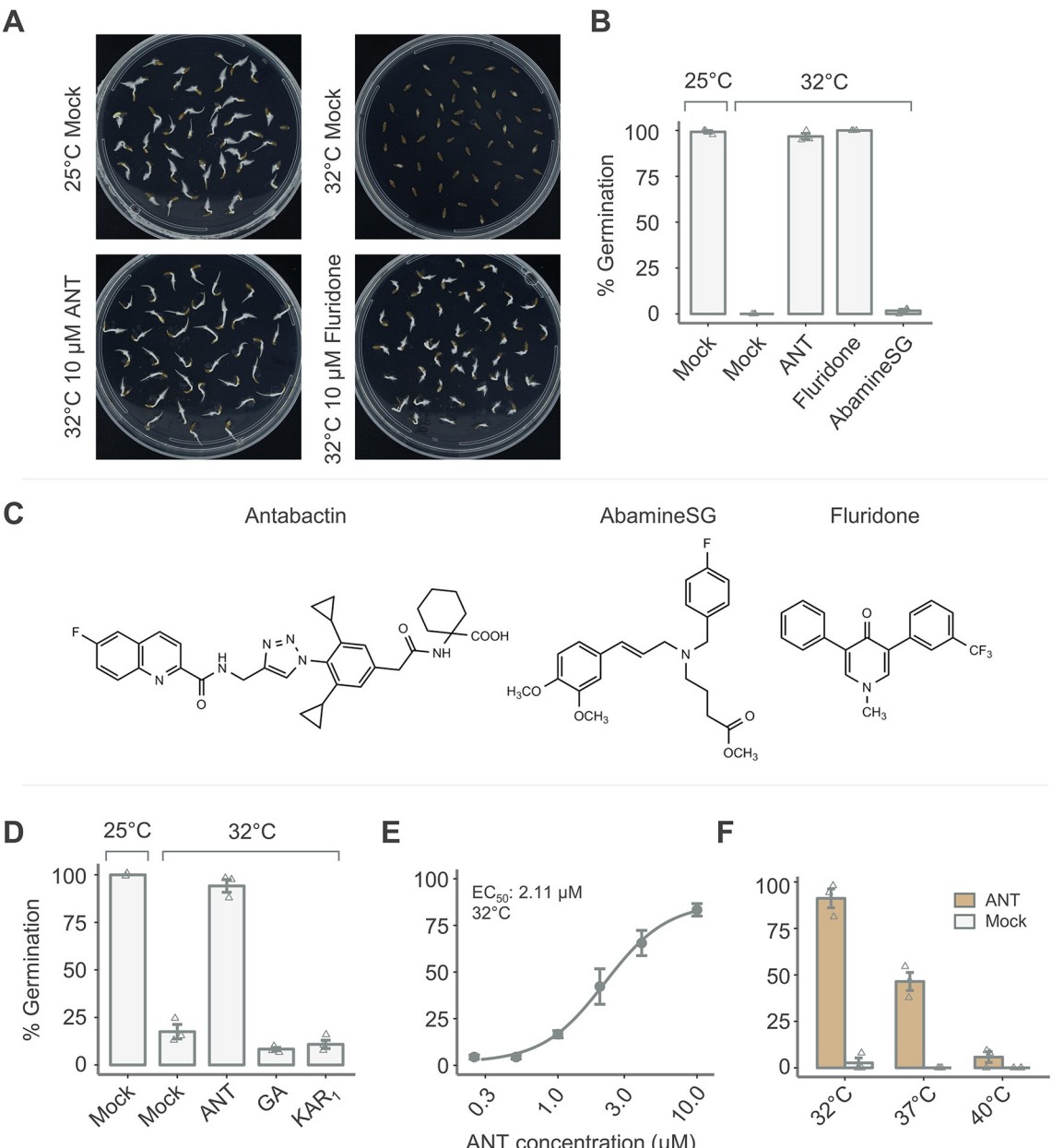

**Fig 1.** ANT improves *L. sativa* 'Salinas' germination under thermoinhibition (A) ANT and fluridone reverse thermoinhibition in *L. sativa* 'Salinas'. Representative images are shown. (B) Barplot summarizing germination. Germination was quantified for seeds imbibed on plates containing DMSO (mock-treated), 10 μM ANT, 10 μM fluridone or 100 μM AbamineSG (representative image in S1 Fig) and grown for 48h. (C) Structures of antabactin, abamineSG and fluridone. (D) GA and KAR1 did not reverse thermoinhibition. Germination was quantified for seeds imbibed on plates containing DMSO, 10 μM ANT, 100 μM GA or 100nM KAR1 and grown for 48h. (E) ANT is potent in the low micromolar range. Germination was quantified for seeds after growth on plates for 24h at 32°C. $ET_{50}$ value inset in the graph are inferred by nonlinear fits in drc. (F) 10 μM ANT reverses thermoinhibition at temperatures of up to 40°C. Germination was quantified seeds after growth in either ANT or mock (DMSO) for 48h. In (A), (B), (D), (E), and (F), liquid sterilized seeds were grown on ½ MS, 0.7% agar plates in the dark. Error bars indicate SEM (n = 3).

viable at 120 hours, and this was confirmed through tetrazolium staining (S3 Fig). These results demonstrate the importance of ABA signaling in germination at high temperatures and show that ANT can reduce thermoinhibition in the low micromolar range.

## Gene expression

To quantify the effect of ANT on the expression of genes associated with thermoinhibition and ABA signaling, we analyzed LsM3K18A, a lettuce homolog of MAPKKK18, a well-characterized ABA response marker gene from Arabidopsis, and the expression levels of LsNCED4, an ABA biosynthetic gene previously implicated in lettuce seed thermoinhibition [7, 23]. First, we wanted to confirm that ANT blocks ABA-mediated transcriptional responses in lettuce. To facilitate this, we identified MAPKKK18 homologs that we named LsM3K18A, B, and C, which phylogenetic analyses show are closely related to AtMAPKKK18 and AtMAPKKK17 (Fig 2A). We imbibed lettuce leaves in mock, ABA and ABA + ANT treatments, extracted RNA and quantified LsM3K18A response through qRT-PCR analyses. LsM3K18A was significantly upregulated (9.5 fold increase) in response to exogenous ABA treatment (100 μM), and this is blocked by a coapplication of ANT (50 μM) and ABA (100 μM) (Fig 2B).

Previous research by Argyris et al. 2008 identified a number of thermoinhibition-responsive genes, one of which was LsNCED4, an ABA biosynthetic gene [7]. LsNCED4 is upregulated during thermoinhibition, and we wanted to determine if ANT could reverse this upregulation. We imbibed mock-treated lettuce seeds at 25˚C and mock and ANT treated seeds at 32˚C, extracted RNA and quantified LsNCED4 response through qRT-PCR analyses. At 32˚C in the dark after 24 hours, it was upregulated 118-fold while at the same temperature LsNCED4 in

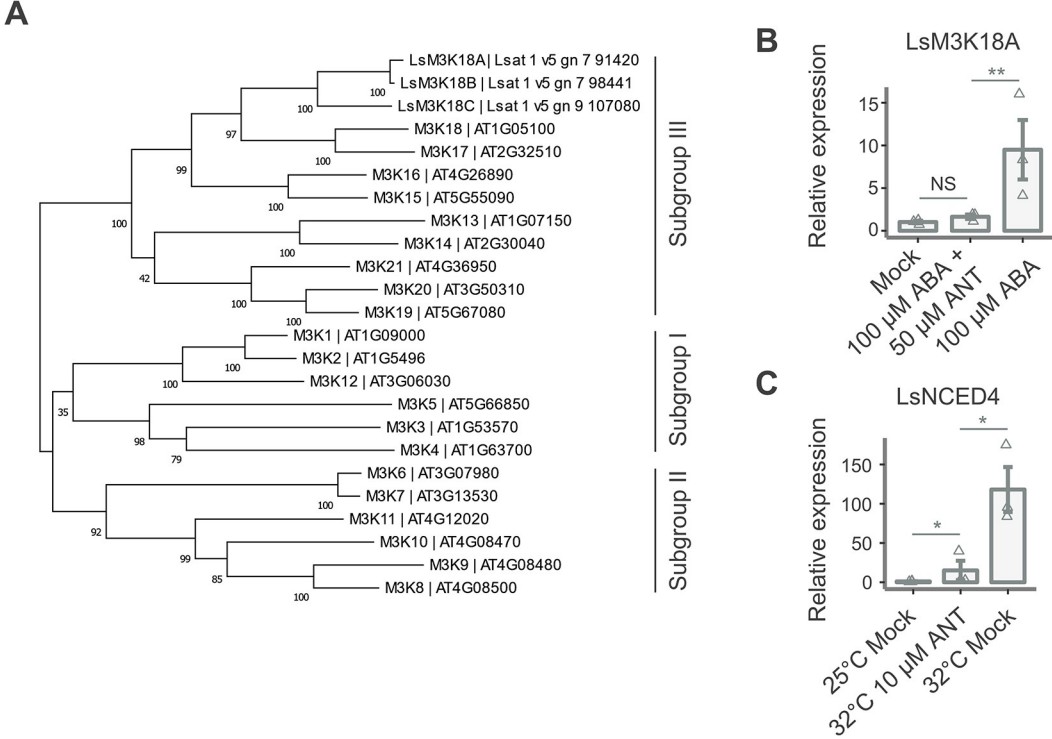

**Fig 2. ANT alters the expression of thermoinhibition and ABA-regulated genes.** (A) Tree inferred using the Maximum Likelihood method and the Tamura-Nei model showing the three subgroups of MAP3Ks in Arabidopsis and the three lettuce homologs of AtMAP3K18 identified by BLAST. Bootstrap support values, derived from 500 replicate analyses, are displayed on the tree to indicate the reliability of each branch. (B) ANT treatment reduces the LsM3K18A expression (normalized to UBC21) measured by qRT-PCR of lettuce leaves in (DMSO) or 100 μM ABA or 100 μM ABA and 50 μM ANT in water. (C) ANT treatment reduces LsNCED4 expression (normalized to UBC21) measured by qRT-PCR of seeds grown for 24 hours in mock (DMSO) or 10 μM ANT. Liquid sterilized seeds were grown in ½ MS, 0.7% agar plates in the dark. * indicates p < 0.05, ** indicates p < 0.01 (pairwise two sample t-test corrected for multiple comparisons). Error bars indicate SEM (n = 3).

ANT-treated seeds was only upregulated 15-fold over mock at 25˚C (Fig 2C). The increased expression under thermoinhibition was consistent with earlier studies and the ANT treatments block the thermoinhbition-associated increase in LsNCED4 transcript levels. We were not able to show any upregulation of LsNCED4 through exogenous ABA application in leaf tissue (S4 Fig). However, LsNCED4 expression had only previously been shown to increase under thermoinhibition with no data on ABA upregulation in lettuce. As such our data for LsNCED4 is consistent with previous experiments in lettuce. Taken together, these analyses show that ANT modulates the expression of key genes involved in ABA signaling, LsM3K18A, and thermoinhibition, LsNCED4, and identified LsM3K18A as an ABA-responsive gene marker gene in lettuce.

## Seed priming

We next explored the potential of delivering ANT into seeds through osmotic priming to assess whether including ANT in the priming process could have an additive effect. We osmotically primed seeds in PEG8000 with and without ANT to deliver ANT into seeds to investigate the uptake and germination effects. Osmotic priming with ANT leads to ANT uptake and storage within lettuce seeds. Three different batches of seeds primed in a solution of 10 μM ANT and -1.25mPa PEG showed they contained ANT (Fig 3A). Seeds were primed, dried, and stored for 5–7 days before extraction and ANT level quantification and all showed ANT similar to seeds primed with only PEG and later spiked with 500nM ANT before LC-MS analysis. The three primed seed batches showed an average of 30% higher germination rates when primed with ANT than solely PEG at 35˚C after 24 hours (Fig 3B and 3C). The seeds that were stored (after-ripened) the longest after harvest had the highest germination under both ANT and mock, 90% and 52% respectively; seeds with the shortest after ripening, 1.5 months, had the lowest germination 30% and 9%. This suggests that priming seeds with ANT has an

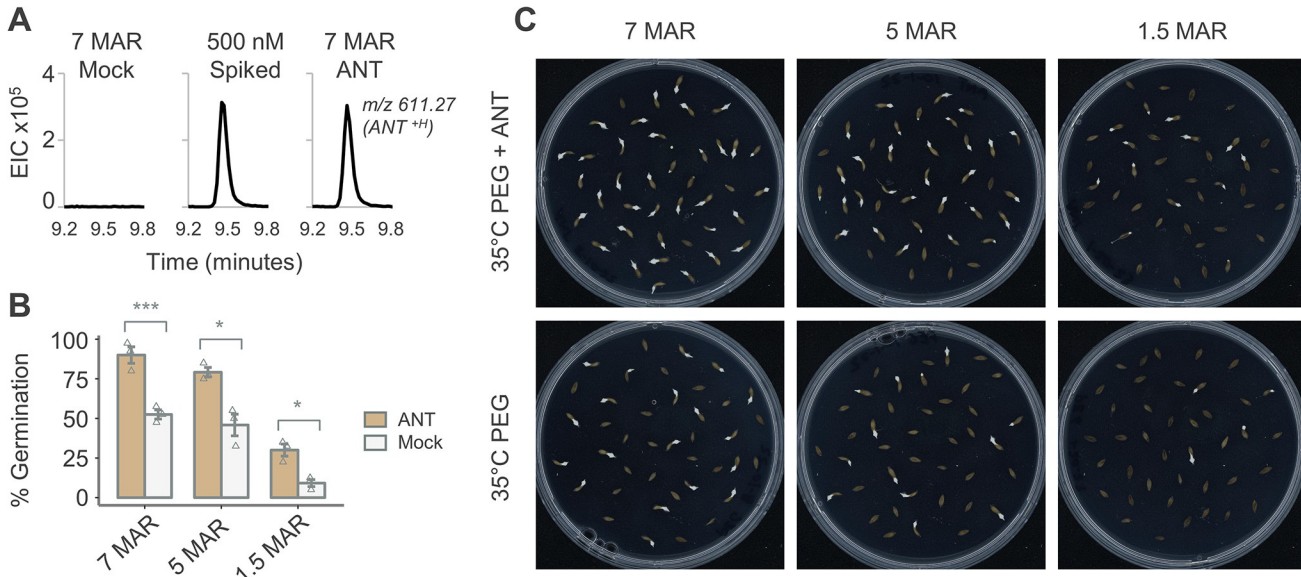

**Fig 3.** Priming with ANT improves lettuce seed germination (A) Extracted ion chromatograms (EIC) for ANT$^{+H}$ m/z 611.270 (+/- 50 ppm). (B) Priming seeds with ANT improves germination rates under thermoinhibition. Germination was quantified for seeds after ripening for 7, 5, and 1.5 months (MAR) primed in -1.25 Mpa solution of PEG 8000 and either 100 μM ANT or mock (DMSO). Primed seeds were plated on ½ MS, 0.7% agar plates, and germination was quantified at 24h at 35˚C. *** indicates p < 0.01 (two-sample t-tests). Error bars indicate SEM (n = 3). (C) Representative images are shown.

additive effect with priming and has the potential to improve germination rates under thermoinhibition.

## Discussion

The phytohormone abscisic acid (ABA) is a critical regulator of thermoinhibition [5, 8, 24]. Lettuce seeds are thermoinhibited at temperatures below the limit for healthy seedling growth, and ABA is an important regulator of this thermoinhibition [3, 8]. Exogenous ABA can reduce the upper limit for germination, and upon thermoinhibition, endogenous ABA levels increase [7].

Herein, we use a new tool to validate ABA's regulation of thermoinhibition by directly blocking ABA signaling with the ABA antagonist antabactin (ANT). To understand the extent of ABA's regulation of germination, we compared it to other chemicals known to influence germination. The phytohormone GA promotes seed germination. In part, it does this by negatively affecting ABA biosynthesis and signaling and promoting ABA catabolism. Argyris et al. 2008 showed that the GA effect of germination in lettuce seeds was variety-dependent, with *L. sativa* 'Salinas' being less affected by GA application [7]. This trend remained consistent in my experiments, where GA did not improve germination under thermoinhibition. Karrikins have also been shown to improve germination in lettuce [25]. Karrikins are chemicals in smoke that can enhance germination, and in *L. sativa* 'Grand Rapids', they have improved germination rates under adverse conditions, specifically after a far-red pulse. However, karrikins have not been tested to reduce the effects of thermoinhibition. In *L. sativa* 'Salinas', we showed that $KAR_1$ treatment did not improve germination rates under temperatures that induced thermoinhibition. While there could be various effects, with 'Salinas' showing less $KAR_1$ response than 'Grand Rapids', these results suggest lettuce seed thermoinhibition is not influenced by karrikins. These results collectively suggest that GA and karrikins are not the central regulators of thermoinhibition.

Previous research has implicated ABA as the primary regulator of thermoinhibition. Chemically, this was demonstrated with fluridone, an ABA biosynthetic inhibitor. Fluridone is a carotenoid biosynthetic inhibitor. It acts by inhibiting phytoene desaturase, which converts phytoene to phytofluene, an early precursor of ABA [26, 27]. Fluridone application results in reduced *de novo* ABA biosynthesis but is a nonspecific inhibitor as it also reduces the synthesis of many other carotenoids, which are often signaling molecules in plants, including strigolactone, another phytohormone [28, 29]. Additionally, blocking carotenoid biosynthesis has an herbicidal effect by removing the photoprotective carotenoids. Despite these disadvantages, it is the most widely used ABA biosynthetic inhibitor. In many species, fluridone promotes and accelerates germination under thermoinhibition. This is also true of lettuce, where applying fluridone promotes germination and reduces the ABA content in imbibed seeds [7, 28]. Although fluridone has some application as an inhibitor in research, its herbicidal effects negate any use in agriculture. While cumulative evidence suggests that ABA is the central regulator of thermoinhibition in lettuce, it depends on indirect evidence, highlighting the need for direct evidence of the regulatory role of ABA in thermoinhibition.

We directly confirmed the central role of ABA signaling in thermoinhibition by blocking ABA signaling with ANT. We applied ANT to thermoinhibited lettuce seeds, and it fully restored germination. This provides direct evidence that ABA is the primary regulator of thermoinhibition in lettuce, suggesting a clear target for crop improvement. We previously showed that ANT also reduces time to germination in heat-treated Arabidopsis seeds, directly demonstrating ABA regulates thermoinhibition in multiple species and that ANT is a valuable tool to assess this across species [17]. Previous research has also suggested that the NCEDs are the rate-limiting step in ABA biosynthesis. Abamine is an NCED inhibitor and reduces ABA

accumulation in planta and promotes seed germination; however, it is phytotoxic [30, 31]. More recently, AbamineSG was developed and demonstrated reduced phytotoxicity and led to reduced ABA content by NCED inhibition [32, 33]. In our experiments, AbamineSG application did not improve germination under thermoinhibition, likely due to ineffective action rather than the NCEDs not being important.

To further confirm ANT's role in inhibiting ABA signaling, we quantified ABA and thermoinhibition-regulated gene transcript levels. ANT reduces thermoinhibition by blocking ABA signaling, affecting the expression of ABA-regulated genes. Specifically, the ABA marker gene M3K18 in Arabidopsis, which is upregulated by ABA or osmotic stress, returns to normal levels following ANT treatment. M3K18 shares a close sequence and transcriptional profile with M3K17, and both genes are functionally redundant [34]. In lettuce, the homologous gene LsM3K18A demonstrated increased expression upon ABA treatment, which ANT treatment subsequently reduced to levels comparable to the control. This indicates LsM3K18A as a useful ABA marker gene in lettuce. LsNCED4 is a well-known factor in influencing thermoinhibition in lettuce [7, 11, 35]. LsNCED4 is the homolog of AtNCED6 [36]. The NCEDs catalyzes a rate-limiting step in ABA biosynthesis; the oxidative cleavage of 9-cis vioxanthin or neoxanthin to xanthoxin, thus, loss of function of the NCEDs results in reduced ABA biosynthesis. This is also the first committed step in ABA biosynthesis [32, 37]. Deletions in LsNCED4 were shown to dramatically increase the maximum temperature at which lettuce could germinate [11]. This directly implicates ABA biosynthesis, and thus, levels and signaling lead to thermoinhibition. Expression of LsNCED4 was previously shown to increase during thermoinhibition, and we found similar trends (Fig 2C) [7]. ANT application, which blocks ABA signaling, reduces expression of LsNCED4. Both LsM3K18A and LsNCED4 expression patterns further demonstrate that ANT's action results from transcriptional regulation of ABA and thermoinhibition-regulated genes.

Transcriptional and chemical approaches have shown that ANT acts through blocking ABA signaling to reduce germination thermoinhibition. We next wanted to quantify the temperature range at which ABA signaling was responsible for thermoinhibition. Argyris et al. 2008 showed that germination of *L. sativa* 'Salinas' decreases at 27°C with germination being fully thermoinhibited at 29°C [9]. Thermoinhibition affects a broad range of crisphead lettuce varieties, as demonstrated by Lafta and Mou (2013). Their study found that, when germinating all tested crisphead lettuce varieties at 34°C, none exhibited a germination rate higher than 80%, with the majority not exceeding 50% germination. To determine whether ANT could improve germination at high temperatures, we first calculated an $EC_{50}$ for ANT at 2.11 μM at 32°C, and, at 10 μM concentrations, germination is fully restored. As such, we used 10 μM ANT for further germination tests. We wanted to determine the extent to which ABA contributes to thermoinhibition even at very high temperatures. Blocking ABA signaling at temperatures of 40°C results in 74% germination. This shows that ABA is the major contributor to thermoinhibition in lettuce, even at very high temperatures. However, upon restoration to normal temperatures, 100% of seeds germinated, suggesting one of two things. Either the ABA effect is not the only factor contributing to thermoinhibition or ANT is not fully blocking ABA signaling. Of note is that germination did slow under 37°C and 40°C, and while at 32°C nearly 100% of seeds were germinated at 48 hours, it took 120 hours before 40°C germination reached 74%. Temperatures of 45°C killed seeds, also suggesting that ABA is the major contributor to thermoinhibition up to the point where seeds are no longer viable.

In agriculture, final emergence rate and synchrony of germination are important factors in determining final crop yield, and, as ABA is the major contributor to thermoinhibition, we wanted to determine if there could be an agriculturally relevant way to deliver ANT to seeds to improve emergence. Previously, targeting NCEDs through CRISPR has been discussed as a way to target ABA-induced thermoinhibition, and it is successful in restoring germination.

However, this method has its challenges [11]. Firstly, gene editing is regulated, and this process must be done to every variety of lettuce grown to produce the desired effect. Secondly, lettuce is grown in a hot climate, and the NCEDs are important parts of the ABA biosynthetic pathway. ABA controls stomatal aperture and thus transpiration, and altering these processes may harm lettuce plants later in their growth [18]. Pre-germination seed treatments can mitigate these issues. As such, we osmotically primed lettuce seeds in PEG and ANT to improve their germination under thermoinhibition. Priming has already been shown to improve germination traits [3, 15, 16]. Additionally, hormonal priming with ABA reduces germination, suggesting that a hormonal priming treatment targeting the ABA pathway can have lasting effects [38]. Our priming treatment successfully delivered ANT into seeds, as shown by LC-MS. Spiking mock-primed seeds with 500nM ANT showed the same peak area as the ANT-primed seeds, suggesting that the concentration of ANT in primed seeds was at least somewhat similar to the 500nM concentration. Priming with ANT and PEG over just PEG promoted higher germination rates and could be beneficial in an agricultural setting by greatly reducing ABA signaling. ANT priming has the potential to be applied to other crops where ABA signaling blocks germination. In light of increased global food demand and the variety of abiotic and biotic factors challenging food production, crop production will need to expand to less suitable growing environments. As this production expands, so will the challenges facing growers. ANT provides another tool to meet these growing challenges.

## Supporting information

**S1 Fig. ANT and fluridone reverse thermoinhibition in *L. sativa* 'Salinas' while abamineSG has no effect.** Germination was quantified for seeds imbibed on plates containing DMSO (mock-treated), 10 μM ANT, 10 μM fluridone or 100 μM AbamineSG and grown for 48h. Representative images are shown.
(TIF)

**S2 Fig. ANT treatment restores germination at temperatures of up to 40˚C.** (A) Seeds treated with 10 μM ANT begin to germinate at temperatures of up to 40˚C after 12048 hours of imbibition. (B) After 120 hours of heat treatment and 72 hours of recovery in dark at 25˚C, seeds germinate.
(TIF)

**S3 Fig. *L. sativa* 'Salinas' seeds are no longer viable at 45˚C.** Seeds were imbibed in dark in 10 μM ANT or mock treatment at 45˚C for 120 hours before tetrazolium staining. These seeds were compared to mock-treated seeds grown in the dark for 24 hours before staining.
(TIF)

**S4 Fig. ABA treatment does not significantly increase LsNCED4 expression.** ABA and ABA + ANT treatments do not alter LsNCED4 expression (normalized to UBC21) measured by qRT-PCR of lettuce leaves in (DMSO) or 100 μM ABA or 100 μM ABA and 50 μM ANT in water (pairwise t test, $p > 0.05$).
(TIF)

**S1 Data. Raw data for all figures.**
(XLSX)

## Acknowledgments

The authors wish to express their gratitude to David Nelson for his generous donation of $KAR_1$, and Tadao Asami for his generous donation of AbamineSG.

## Author Contributions

**Conceptualization:** James Eckhardt, Aditya Vaidya.

**Funding acquisition:** Sean Cutler.

**Investigation:** James Eckhardt.

**Methodology:** James Eckhardt.

**Resources:** Aditya Vaidya.

**Supervision:** Sean Cutler.

**Visualization:** James Eckhardt.

**Writing – original draft:** James Eckhardt.

**Writing – review & editing:** Sean Cutler.

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
