## [Decision Letter · Decision Letter 0]

18 Jun 2024

PONE-D-24-20564The ABA receptor antagonist antabactin restores the germination of thermoinhibited lettuce seedsPLOS ONE

Dear Dr. Cutler,

Thank you for submitting your manuscript to PLOS ONE. After careful consideration, we feel that it has merit but does not fully meet PLOS ONE’s publication criteria as it currently stands. Therefore, we invite you to submit a revised version of the manuscript that addresses the points raised during the review process.

Your manuscript was reviewed by two experts in the field. The reviewers found the work interesting but raised several issues which need to be addressed properly. The reviewers provide detailed comments in their reviews and point out the areas where the manuscript needs to be improved. I also read the manuscript carefully and largely agree with the reviewers’ comments.

We look forward to receiving your revised manuscript.

Kind regards,

Mohammad Irfan, Ph.D.

Academic Editor

PLOS ONE

Journal Requirements:

"This work was supported by NSF IOS grant 1656890."

"The authors wish to express their gratitude to David Nelson for his generous donation of KAR1, and Tadao Asami for his generous donation of AbamineSG. This work was partially supported by NSF grant 1656890."

"The authors wish to express their gratitude to David Nelson for his generous donation of KAR1, and Tadao Asami for his generous donation of AbamineSG. This work was partially supported by NSF grant 1656890."

Reviewers' comments:

Reviewer's Responses to Questions

**Comments to the Author**

1. Is the manuscript technically sound, and do the data support the conclusions?

Reviewer #1: Partly

Reviewer #2: Yes

2. Has the statistical analysis been performed appropriately and rigorously? 

Reviewer #1: I Don't Know

Reviewer #2: Yes

3. Have the authors made all data underlying the findings in their manuscript fully available?

Reviewer #1: Yes

Reviewer #2: Yes

4. Is the manuscript presented in an intelligible fashion and written in standard English?

Reviewer #1: No

Reviewer #2: Yes

5. Review Comments to the Author

Reviewer #1: The article is on understanding dormancy in lettuce seeds. The authors studied ABA and ANT to investigate if it can mitigate the effects of dormancy and high temperatures on seed germination. The authors looked into many aspects including gene expression. The article needs to be rewritten as it lacks scientific coherency.

• Introduction is very general and incoherent. Many sentences seem out of place and lack references. Please rewrite the introduction and make it more scientific. The problem is not introduced properly, and many different aspects are discussed without providing proper context. Greatly recommend to rewrite introduction and make it less vague. ANT is also not properly discussed. For an improved structure, consider starting with dormancy and its importance in agriculture, then the crops that require it. Later move on to how is it regulated, then hormones, then regulators. Then go to examples. Discuss ANT and previous work done on ANT.

• Make sure to correctly explain the relation between ANT, ABA and MAP kinase. Its very vague in introduction.

• The writing is not up to the mark. Greatly recommend that this paper should be reviewed thoroughly. The presentation is weak and not coherent.

• In methods, please provide the conditions in which the Seeds of Lactuca sativa ‘Salinas’ were grown in a greenhouse.

• It is not clear from the methods is what tissue was used for gene expression study. If seeds were grown in dark for only 24 hours, was RNA extracted from seed? Was extracted from endosperm or radicle or cotyledon. Please explain the right procedure used.

• Why 50 mg for seeds (considering seed was used) and 100 mg for leaf tissue for RNA extraction.

• Not much information was provided oh how the gene expression analysis. I recommend reading MIQE guidelines and provide as many details as possible.

• Why were ABA and ANT treatments not conducted on live plants. Why was detached and cut leaf was used. Once leaves or any tissues are cut the physiology changes. How does it represent true gene expression.

• Were tissues snap frozen in liquid nitrogen before RNA extraction?

• Only UBC21 was used as reference gene for normalization. However, the current standard is to use at least two reference genes. Pleas explain.

• Many sentences in the results should have been part of discussion. In results only the results should be mentioned.

Lines 51 – 52: Please elaborate more and add a reference.

Lines 51 – 59: The paragraph is incoherent and general. Please rephrase to add clarity.

Lines 67 – 69: The above paragraph is a discussion about hormones and (DOG1) a regulator is discussed without a context. Seems out of place.

Line 127: Just curious why was DMSO used as control instead of DI water.

Lines 123 – 129: Is this method of ANT treatment based on previously reported methods? if so, please provide references.

Lines 156: What two tissue types were used ?

Lines 155 – 171: Very vague. Please rewrite.

Lines 203 – 207: This should be part of the discussion.

Lines 203 – 219: The result section is not very clear. Please explain the inference drawn coherently.

Lines 247 – 249: This inference drawn is not accurate. It only shows ANT has effect on germination. How did you conclude ABA signaling was involved? This should be part of discussion not results.

Reviewer #2: In the manuscript titled “The ABA receptor antagonist antabactin restores the germination of thermoinhibited lettuce seeds,” the authors have done an excellent work in understanding the molecular mechanism and solving the problem of seed vigor in lettuce. This work is directly applicable to the agriculture of lettuce and may be further extended to other commercial crops. I have the followings comments and recommendations:

Comments:

1. Figure 1A contains a picture plate, a graph, and the molecular structure of the chemical. Divide these into sub-figures. Nothing is mentioned about the molecule's chemical structure in the text or the figure legend. The authors should mention it in both places.

2. Line 208: Mention both pictures and graphs in-text for readers to easily refer to the data discussed.

3. Line 210: (…32 C?) Remove the question mark and correct the symbol from oC to oC.

4. Line 211, 212, 216: Refer to the figure panel discussed. Make sure that the exact figures are mentioned wherever the data is discussed in-text throughout the manuscript.

5. Line 223: Include a picture plate for the treatment of seeds with AbamineSG similar to the pictures shown for treatment with Fluridone and ANT.

6. Line 243: The correct figure is not cited in this section. Fig S3 and Fig 1D seem interchanged.

7. The color code for the graph in Fig S3 is missing.

8. Line 242-249: Reorganize the data discussed in this section. The data discussed in this section is represented in Fig 1D, Fig S1, and Fig S3. The ordering of figures should match the order of the discussion. Preferably club the two supplementary figures. Also, the values in line 243 and line 244 do not match the graph shown.

9. The authors check the change in expression of LsM3K18A using qRT-PCR. Did they check the expression pattern of 18B and C and was there a reason why only 18A was selected and not 18B and C. I understand that leaves could be used to understand the transcriptional response of ABA but using the seed tissue would give a better picture. The author can use the same RNA as the one used for NCED4 expression analysis to also see the gene expression change of LsM3Ks.

10. Figure 2B is discussed later in the text than 2C. The order of figures should match the order of the in-text discussion. Check throughout the text.

11. The author should show the expression pattern of other NCEDs during the thermoinhibition of lettuce seeds.

12. Line 259: Delete (Lsat_1 …..)

13. Line 274: replace out with “our”

14. Divide picture plate and graphs into different subfigure in fig 3. Figure order …..

15. Mention color code for the graph in Figure 3A

16. Line 305: “ the seeds …after-ripened the longest” this might confuse the reader. Author can use something similar to “seeds stored for longer duration since ripening”

6. PLOS authors have the option to publish the peer review history of their article (what does this mean?). If published, this will include your full peer review and any attached files.

Reviewer #1: No

Reviewer #2: No

---

## [Author Response · Author response to Decision Letter 0]

5 Sep 2024

We have included a separate file that details our responses to the reviewers. Here it is again if needed here, too:

Response to reviewers

Academic editor

We ensured that the document meets the requirements.

"This work was supported by NSF IOS grant 1656890."

We have provided this in the cover letter.

"The authors wish to express their gratitude to David Nelson for his generous donation of KAR1, and Tadao Asami for his generous donation of AbamineSG. This work was partially supported by NSF grant 1656890."

"The authors wish to express their gratitude to David Nelson for his generous donation of KAR1, and Tadao Asami for his generous donation of AbamineSG. This work was partially supported by NSF grant 1656890."

This has been amended in the manuscript and added to the cover letter.

Reviewer #1: The article is on understanding dormancy in lettuce seeds. The authors studied ABA and ANT to investigate if it can mitigate the effects of dormancy and high temperatures on seed germination. The authors looked into many aspects including gene expression. The article needs to be rewritten as it lacks scientific coherency.

• Introduction is very general and incoherent. Many sentences seem out of place and lack references. Please rewrite the introduction and make it more scientific. The problem is not introduced properly, and many different aspects are discussed without providing proper context. Greatly recommend to rewrite introduction and make it less vague. ANT is also not properly discussed. For an improved structure, consider starting with dormancy and its importance in agriculture, then the crops that require it. Later move on to how is it regulated, then hormones, then regulators. Then go to examples. Discuss ANT and previous work done on ANT.

1. We have extensively rewritten the introduction to better align its narrative with the rest of the work; in hindsight, we can see that the original focus on seed dormancy distracted from the major points of the manuscript which are now better articulated. Thank you for helping us improve the manuscript. 

• Make sure to correctly explain the relation between ANT, ABA and MAP kinase. Its very vague in introduction.

2. This has been clarified in the introduction.

• The writing is not up to the mark. Greatly recommend that this paper should be reviewed thoroughly. The presentation is weak and not coherent.

3. We have extensively revised the introduction, results, and discussion to improve clarity.

• In methods, please provide the conditions in which the Seeds of Lactuca sativa ‘Salinas’ were grown in a greenhouse.

4. We have included more details on growing conditions in the methods section.

• It is not clear from the methods is what tissue was used for gene expression study. If seeds were grown in dark for only 24 hours, was RNA extracted from seed? Was extracted from endosperm or radicle or cotyledon. Please explain the right procedure used.

5. Whole seeds. This has been added to the methods.

• Why 50 mg for seeds (considering seed was used) and 100 mg for leaf tissue for RNA extraction.

6. Seeds are a rich RNA source and required less material; we have noted this in the methods section

• Not much information was provided oh how the gene expression analysis. I recommend reading MIQE guidelines and provide as many details as possible.

7. We have added additional information on how the gene expression analysis was performed. RNA extraction, reverse transcription and qPCR was performed using kits according to the manufacturers recommendations. We have now noted this and any alterations to the protocols that were made.

• Why were ABA and ANT treatments not conducted on live plants. Why was detached and cut leaf was used. Once leaves or any tissues are cut the physiology changes. How does it represent true gene expression.

8. We used leaf discs because this is frequently employed in ABA-response studies and allows for efficient ABA uptake, which can be problematic in plants like lettuce with more pronounced cuticles. The primary goal of our qPCR experiments was to establish ANT’s anti-ABA activity, which this experiment demonstrates. We have modified the relevant methods section to make the rationale for this experimental choice clear.

• Were tissues snap frozen in liquid nitrogen before RNA extraction?

9. We used a Precellys 24 tissue lyser and Qiagen RNA isolation kit using their recommended procotol (this has now been clarified in the methods and we apologize for the omission). The Qiagen protocol harvested, placed in a lysis buffer, and extracted using disruption without prior freezing.

• Only UBC21 was used as reference gene for normalization. However, the current standard is to use at least two reference genes. Please explain.

10. We have previously characterized ANT’s effects in Arabidopsis and wheat. Our data showed that ANT is a potent blocker of cellular ABA responses that can reduce ABA-induced expression of target genes by ~ 10 - 20x, depending on the marker gene and experimental conditions (Vaidya et al. 2021). Given this context, the primary goal of our qRT-PCR experiments was to ensure that ANT possesses antagonist activity in lettuce, as expected based on our work in wheat and Arabidopsis. Indeed, our qRT-PCR data in the current manuscript demonstrate ~10-fold and ~100x reductions in the levels of two different ABA pathway genes after exogenous ABA or heat treatments (which triggers ABA biosynthesis and ABA-dependent thermoinhibition as per prior data by Argyris et al. 2008). Although it would be nice to have more reference genes, the data provided strongly support the conclusion that ANT antagonizes ABA responses in lettuce, as expected based on prior characterization of its effects in both monocots and dicots.

• Many sentences in the results should have been part of discussion. In results only the results should be mentioned.

11. We have moved several sentences from the results section to the discussion

Lines 51 – 52: Please elaborate more and add a reference.

12. We have extensively rewritten the introduction; this paragraph has been removed.

Lines 51 – 59: The paragraph is incoherent and general. Please rephrase to add clarity.

13. We agree. See our previous response above.

Lines 67 – 69: The above paragraph is a discussion about hormones and (DOG1) a regulator is discussed without a context. Seems out of place.

14. We agree. See our previous response above.

Line 127: Just curious why was DMSO used as control instead of DI water.

15. DMSO is the carrier solvent we use to solvate ANT and all other chemicals used in the paper. We have noted this in the methods and apologize for this omission. All experiments contained 0.1% vol/vol DMSO and differing concentrations of test compounds.

Lines 123 – 129: Is this method of ANT treatment based on previously reported methods? if so, please provide references.

16. Yes, the methods were adapted from Vaidya et al. 2021 where we did something similar; we have noted this in the text

Lines 156: What two tissue types were used ?

17. Seeds and leaves (we have noted this in the revised methods)

Lines 155 – 171: Very vague. Please rewrite.

18. We have updated this and provided additional information. 

Lines 203 – 207: This should be part of the discussion.

19. This section was removed from the results and we have moved this section to the introduction.

Lines 203 – 219: The result section is not very clear. Please explain the inference drawn coherently.

20. This section has been revised to improve clarity.

Lines 247 – 249: This inference drawn is not accurate. It only shows ANT has effect on germination. How did you conclude ABA signaling was involved? This should be part of discussion not results.

21. We believe this is a reasonable conclusion given that ANT is a biochemically well-characterized pan-ABA receptor antagonist with pM binding affinity that blocks ABA responses in Arabidopsis, tomato, barley, and, as we show here in lettuce. However, we have modified the wording to improve precision.

Reviewer #2: In the manuscript titled “The ABA receptor antagonist antabactin restores the germination of thermoinhibited lettuce seeds,” the authors have done an excellent work in understanding the molecular mechanism and solving the problem of seed vigor in lettuce. This work is directly applicable to the agriculture of lettuce and may be further extended to other commercial crops. I have the followings comments and recommendations:

Comments:

1. Figure 1A contains a picture plate, a graph, and the molecular structure of the chemical. Divide these into sub-figures. Nothing is mentioned about the molecule's chemical structure in the text or the figure legend. The authors should mention it in both places.

We added a pointer from the text to the figure; and modifed the figure labels.

2. Line 208: Mention both pictures and graphs in-text for readers to easily refer to the data discussed.

Added this to the text.

3. Line 210: (…32 C?) Remove the question mark and correct the symbol from oC to oC.

We checked and confirmed the experimental conditions and corrected as recommended.

4. Line 211, 212, 216: Refer to the figure panel discussed. Make sure that the exact figures are mentioned wherever the data is discussed in-text throughout the manuscript.

We clarified and updated the text with exact figure references.

5. Line 223: Include a picture plate for the treatment of seeds with AbamineSG similar to the pictures shown for treatment with Fluridone and ANT.

We have added this image in Figure S1 and referenced it in the Fig 1 caption.

6. Line 243: The correct figure is not cited in this section. Fig S3 and Fig 1D seem interchanged.

This has been corrected in the manuscript.

7. The color code for the graph in Fig S3 is missing.

This has been corrected in the figure.

8. Line 242-249: Reorganize the data discussed in this section. The data discussed in this section is represented in Fig 1D, Fig S1, and Fig S3. The ordering of figures should match the order of the discussion. Preferably club the two supplementary figures. Also, the values in line 243 and line 244 do not match the graph shown.

 We have shifted the naming scheme and order of the supplemental figures to match the order they are discussed in the text. We incorrectly plotted the 48hr data in the supplemental figure, have changed it to reflect the values at 120 hours and apologize for the error. We chose to keep the two supplementary figures separate from one another since they are separate experiments that address different points.

9. The authors check the change in expression of LsM3K18A using qRT-PCR. Did they check the expression pattern of 18B and C and was there a reason why only 18A was selected and not 18B and C. I understand that leaves could be used to understand the transcriptional response of ABA but using the seed tissue would give a better picture. The author can use the same RNA as the one used for NCED4 expression analysis to also see the gene expression change of LsM3Ks.

Our primary goal was to develop a marker to confirm ANT’s efficacy in blocking ABA signaling. We did not analyze MAPKKK18B or MAPKKK18C expression, as MAPKKK18A showed a robust ABA response in leaves and served its purpose of providing this control (see our response to reviewer 1 question 10).

10. Figure 2B is discussed later in the text than 2C. The order of figures should match the order of the in-text discussion. Check throughout the text.

 The order of Fig 2B and C has been switched as have references in the text. The order of Fig 3 was also corrected. The ordering of supplemental figures was also altered to correspond with order referenced in text.

11. The author should show the expression pattern of other NCEDs during the thermoinhibition of lettuce seeds.

 LsNCED1 and LsNCED4 expression data was published previously (Argyris et al., 2008). LsNCED1 had low expression levels and LsNCED4 has been demonstrated as the key NCED that contributes to lettuce seed thermoinhibition and was therefore the focus of our experiments.

12. Line 259: Delete (Lsat_1 …..)

This has been removed and moved to the methods.

13. Line 274: replace out with “our”

 This has been corrected in the manuscript.

14. Divide picture plate and graphs into different subfigure in fig 3. Figure order …..

Both of these items have been corrected.

15. Mention color code for the graph in Figure 3A

 This has been added.

16. Line 305: “ the seeds …after-ripened the longest” this might confuse the reader. Author can use something similar to “seeds stored for longer duration since ripening”

This has been clarified in the text.

---

## [Decision Letter · Decision Letter 1]

24 Sep 2024

PONE-D-24-20564R1The ABA receptor antagonist antabactin restores the germination of thermoinhibited lettuce seedsPLOS ONE

Dear Dr. Cutler,

Thank you for submitting your manuscript to PLOS ONE. After careful consideration, we feel that it has merit but does not fully meet PLOS ONE’s publication criteria as it currently stands. Therefore, we invite you to submit a revised version of the manuscript that addresses the points raised during the review process. The reviewer 2 still has a few minor concerns. Please address them appropriately.

We look forward to receiving your revised manuscript.

Kind regards,

Mohammad Irfan, Ph.D.

Academic Editor

PLOS ONE

Journal Requirements:

Reviewers' comments:

Reviewer's Responses to Questions

**Comments to the Author**

1. If the authors have adequately addressed your comments raised in a previous round of review and you feel that this manuscript is now acceptable for publication, you may indicate that here to bypass the “Comments to the Author” section, enter your conflict of interest statement in the “Confidential to Editor” section, and submit your "Accept" recommendation.

Reviewer #1: All comments have been addressed

Reviewer #2: All comments have been addressed

2. Is the manuscript technically sound, and do the data support the conclusions?

Reviewer #1: Yes

Reviewer #2: Yes

3. Has the statistical analysis been performed appropriately and rigorously? 

Reviewer #1: I Don't Know

Reviewer #2: Yes

4. Have the authors made all data underlying the findings in their manuscript fully available?

Reviewer #1: Yes

Reviewer #2: Yes

5. Is the manuscript presented in an intelligible fashion and written in standard English?

Reviewer #1: Yes

Reviewer #2: Yes

6. Review Comments to the Author

Reviewer #1: Thank you for addressing my concerns. All my comments are addressed. I have recommended to accept The ABA receptor antagonist antabactin restores the germination of thermoinhibited

lettuce seeds

Reviewer #2: I am happy with the edits made to the manuscript.

I have few minor recommendations for the manuscript:

1. Line 65-69: Too long statement. Make shorter statements and use proper punctuation for more clarity.

2. Line 81: replace "; it" with "which"

3. Italicize all scientific names

4. Methods: Gene expression assay: Mention if extracted RNA was treated for the removal of genomic DNA contamination.

5. Capitalize each word of "Crispr" to "CRISPR" throughout the text.

6. Line 423: change LCMS to LC-MS

7. PLOS authors have the option to publish the peer review history of their article (what does this mean?). If published, this will include your full peer review and any attached files.

Reviewer #1: No

Reviewer #2: No

---

## [Author Response · Author response to Decision Letter 1]

14 Nov 2024

We made the minor grammatical and stylistic changes requested by reviewer 2.

---

## [Editor Report · Decision Letter 2]

25 Nov 2024

Chemical Disruption of ABA Signaling Overcomes High-Temperature Inhibition of Seed Germination and Enhances Seed Priming Responses

PONE-D-24-20564R2

Dear Dr. Cutler,

We’re pleased to inform you that your manuscript has been judged scientifically suitable for publication and will be formally accepted for publication once it meets all outstanding technical requirements.

Kind regards,

Mohammad Irfan, Ph.D.

Academic Editor

PLOS ONE
---

## [Editor Report · Acceptance letter]

2 Dec 2024

PONE-D-24-20564R2 

PLOS ONE

Dear Dr. Cutler, 

I'm pleased to inform you that your manuscript has been deemed suitable for publication in PLOS ONE. Congratulations! Your manuscript is now being handed over to our production team.

Kind regards, 

on behalf of

Dr. Mohammad Irfan 

Academic Editor

PLOS ONE